# Analysis of Training Deep Learning Models for PCB Defect Detection

**DOI:** 10.3390/s23052766

**Published:** 2023-03-02

**Authors:** Joon-Hyung Park, Yeong-Seok Kim, Hwi Seo, Yeong-Jun Cho

**Affiliations:** 1Data Science Team, Hyundai Mobis, 203 Teheran-ro, Gangnam-gu, Seoul 06141, Republic of Korea; 2Department of Artificial Intelligence Convergence, Chonnam National University, 77 Yongbong-ro, Buk-gu, Gwang-ju 61186, Republic of Korea

**Keywords:** defect inspection, deep learning, machine vision, product inspection, smart manufacturing, smart factory

## Abstract

Recently, many companies have introduced automated defect detection methods for defect-free PCB manufacturing. In particular, deep learning-based image understanding methods are very widely used. In this study, we present an analysis of training deep learning models to perform PCB defect detection stably. To this end, we first summarize the characteristics of industrial images, such as PCB images. Then, the factors that can cause changes (contamination and quality degradation) to the image data in the industrial field are analyzed. Subsequently, we organize defect detection methods that can be applied according to the situation and purpose of PCB defect detection. In addition, we review the characteristics of each method in detail. Our experimental results demonstrated the impact of various degradation factors, such as defect detection methods, data quality, and image contamination. Based on our overview of PCB defect detection and experiment results, we present knowledge and guidelines for correct PCB defect detection.

## 1. Introduction

A printed circuit board (PCB) consists of various electronic parts, such as capacitors, resistors, semiconductors, and conductive paths. The PCB performs an electrical connection between the parts through conductive paths on the substrate. Additionally, PCBs simplify the development of prototypes and enable the mass production of electronic products. They are generally used in home electronic products or industrial electronic facilities. As the industry changes, PCBs have also been installed in automobiles and mechanical facilities, where PCBs were previously not essential [1]. In recent years, most electrical, electronic, and mechanical products in our daily lives have contained a variety of PCBs. Thus, accurate PCB production is becoming critical as the demand for these electronic products increases [2].

Accordingly, many companies have employed automated inspection systems for PCB manufacturing [3]. Automatic optical inspection (AOI) systems automatically detect PCB components in terms of their surfaces, arrangements, and circuits. The system takes images for defect detection. (Usually, AOI systems take two-dimensional color images (RGB) but can take other images, such as three-dimensional, ultrasonic, X-ray, and infrared images, depending on the purpose of the system [4,5]. In this study, we focus on two-dimensional RGB images.) For highly reliable PCB manufacturing, defect detection methods embedded in AOI systems should be accurate. However, traditional defect detection methods operate only on rule-based algorithms [6,7], which are vulnerable to contaminated PCB images and have difficulty coping with new types of PCB defects. Recent attempts using deep learning have been studied to overcome the limitations of the rule-base methods, and they display improved defect detection performance compared with conventional methods [8,9,10,11].

However, even if deep learning achieves positive results in PCB defect detection, deep learning models are not perfect for all cases. Extensive training data and various training data distributions are required to train deep neural networks with numerous learning parameters, such as weights and biases. Particularly, the balance of the training data is essential for performing deep learning [12]. A deep learning model is preferred in industrial fields with varied and balanced training data. Industrial images, such as PCB images, are challenging to collect compared with other typical image data [13,14] because PCB images can only be collected during the manufacturing process in the factory. Furthermore, the filming environment of PCB images is different from other standard images, making model training difficult, as noted in Section 3. In this case, introducing deep learning techniques should be more carefully performed for PCB manufacturing.

Recently, many attempts have been studied to improve existing defect detection models or proposed new types of defect detection methodologies for specific industrial sites [15,16]. Despite these attempts, the models can still be degraded by several factors such as a lack of training data, unbalanced training data, image data contamination during system maintenance, and inappropriate model selection (Figure 1b). The application of technology in industrial fields requires considerable cost and time. Therefore, it is essential to review the various performance degradation factors in advance and apply an appropriate model training method.

In this study, we analyze training methodologies for deep learning models that can stably perform PCB defect detection in various industrial situations. The main contributions of this study are as follows: We first summarize the properties of PCB images and review possible degradation factors such as image contamination in industrial sites. Second, we organize applicable defect detection methods according to system purpose and we review the properties of each method in detail. Finally, we present methodologies and guidelines of proper deep learning for defect detection by analyzing various performance degradation factors in real industrial sites. To sum up, this study considers various performance degradation factors while presenting a possible methodology for PCB defect detection. We hope that this study provides meaningful guidelines to readers who desire to apply defect detection methods using deep learning in many industrial fields.

## 2. Related Work

In traditional image processing and computer vision fields, research has focused on extracting and learning discriminating features in images. Many studies have extracted critical features, such as the shape, pattern, and color of images [17,18], to learn high-performance classifiers [19,20]. However, the emergence of deep learning in recent years, which can be optimized with one objective function from feature extraction to the classification stage, has led to a paradigm shift in many research fields [21]. In particular, recent deep neural network structures aimed at image understanding, such as convolutional neural networks, have achieved significant performance improvements that far exceed existing image understanding techniques, regardless of application [22]. Computer vision methods have steadily improved, e.g., image classification, understanding, and object detection methods. These methods have been applied to various fields, including defect detection at industrial sites.

It has long been attempted to detect defects in industrial data using AOI systems. There are two main approaches for automated PCB defect detection: (1) rule-based defect detection, (2) deep learning-based defect detection. In the rule-based approach, many studies have determined defects based on designed rules after performing simple binary image processing on PCB data [3,6,23]. However, these simple analyses have difficulty coping with various situations that may occur in industrial sites. For example, they have difficulty handling subtle image changes due to position and lighting differences [24]. The studies have not adequately reviewed the factors or situations that can potentially degrade the performance of designed algorithms.

Meanwhile, many methods have been studied to perform defect detection via convolutional neural networks to overcome the limitations of these rule-based methods [8,25]. For more efficient applications, Ren et al. proposed a method to use a pre-trained deep learning network on surface datasets that have very few data points and require accurate identification of defect regions [26]. Kim et al. also performed defect detection using an auto-encoder structure [15]. They first generated a nondefective image from a defective image, then compared the two images (defective and nondefective) to perform defect detection rather than learning a detector that directly detects PCB defects. Liao et al. proposed the YOLOv4-MN3 model, a modified version of YOLOv4 that can detect PCB defects quickly and efficiently [16]. In addition, Adibhatla et al. optimized the YOLO structure for improving defect detection accuracy [27]. Lian et al. utilized a mask R-CNN for defect detection and added a geometric attention-guided mask branch into the fully connected CNN for increasing efficiency of the model [28]. As mentioned, studies have attempted various methods for defect detection in specific industrial applications. However, factors that can interfere with learning in the industrial field are very diverse, so these factors should be identified and addressed in advance. While previous studies have focused on network design or learning methods for PCB defect detection, our study instead focuses on various degradation factors that can affect the model’s performance. We aim to review and analyze potential model degradation factors on industrial data, which, to our knowledge, has not been explored in prior research.

## 3. Properties of PCB Images

### 3.1. Public PCB Datasets

Available PCB image data for research purposes are summarized in Table 1. The PCB [29] has synthetic six-class defect images from 10 independent PCB images. Five classes (e.g., mousebite, open, short, spurs, and spurious) are defects of PCB circuits, and the other class (missing hole) is a defect of a PCB component. Each defective class randomly generated 115 PCB images, providing 690 total PCB images, including defects provided in the dataset. The TDD-PCB [30] extends the PCB [29] dataset to over 10,000 images through geometric and photometric image transformations. The augmented image has 600 × 600 pixels and includes two defects, but both datasets (PCB [29] and TDD-PCB [30]) have several limitations. The source PCB images comprise only 10 images, and the defective images are not real images but are generated synthetic data in the datasets [29,30].

In contrast, the FICS-PCB [31], PCB DSLR [32], and PCB-METAL [33] datasets provide a relatively large number of source PCB images compared with PCB and TDD-PCB. However, they do not provide defective PCB parts because defective data are scarce in actual industrial sites, and even if defects occur, companies do not share their defective parts with researchers. In this work, we used TDD-PCB [30] data, which provide various PCB defect images.

### 3.2. Attributes of Industrial Data

Attributes of industrial data in manufacturing sites are summarized in Table 2. According to the work in [34], industrial data are generally not rich in volume and are imbalanced. The PCB image data (TDD-PCB [30]) generated at an industrial site also have the attributes in Table 2. For example, TDD-PCB has only 10 original source images (*small amount*), and the defective parts account for a small proportion of the PCB (*small object*). The dataset has 21,336 defective images, but all other image areas except the defective area can be extracted as nondefective images. In other words, defective and nondefective images are imbalanced, as illustrated in Figure 2a (*imbalanced*). In addition, six types of defective classes exist in TDD-PCB, which can be divided into two main categories: (1) part-related defects (missing hole) and (2) circuit-related defects (mousebite, open, short, spur, and spurious). As demonstrated in Figure 2b, it is difficult to distinguish the circuit-related defects due to the high similarity of appearance (fine-grained). The attributes mentioned above, such as *small amount*, *imbalance*, and *fine-grained*, lead to difficulties in training and applying deep learning models. Training numerous parameters in deep neural networks without sufficient training data can cause over-fitting problems. Similarly, imbalanced data between each class can cause model bias.

Furthermore, the industrial data are likely to be contaminated due to the environment of the manufacturing space (*strong interference*) and the long-term maintenance of facilities (*temporality*). The product manufacturing process involves chemical and physical reactions on boards and parts; thus, illumination changes and blurring can occur in the obtained images. Moreover, although products are not contaminated yet, they can become contaminated when the manufacturing facilities are not adequately maintained. We discuss the possible image contamination of the industrial data in Section 3.3. When analyzing industrial data, such as PCB images, all attributes should be considered before training the models.

### 3.3. Image Contamination

This section examines possible image contamination due to the attributes of industrial data discussed in Section 3.2. Hendrycks et al. [35] propose various image contamination models to evaluate the robustness and stability of deep neural networks. They considered common and industrial image acquisition conditions. Table 3 summarizes the possible contamination according to the image acquisition environment of PCB data. Examples of possible contamination are depicted in Figure 3. We categorized industrial environments for image acquisition according to four factors as follows:**Close-up imaging:** Close-up imaging is required to inspect small electronic parts in detail. However, a small vibration of the camera or product causes significant blurring and image degradation. For example, subtle changes in intrinsic camera parameters occur, such as defocus and zoom blurring. In addition, subtle camera pose changes or PCB movements occur, such as elastic transforms and motion blur.**Illumination changes:** Generally, manufacturing equipment blocks external light and uses internal lighting to avoid the influence of external light sources. The quality of the image changes sensitively depending on the lighting condition. According to the lighting variation, images are affected by noise (e.g., Gaussian and impulse) and color variations (e.g., brightness change, contrast change, and saturation).**Long-term maintenance:** During long-term manufacturing, various image degradation factors, such as dirt on the lens and PCB, dust and steam particles in the air, and others, occur, which can cause glass blur and spatter contamination.**Systematic issues:** Factory facilities require enough storage, but some older factories may not be equipped with enough storage capacity to handle the generation of vast amounts of image data. For example, multiple high-resolution images (e.g., 4 K image) are taken for each product and thousands of products are produced per day, a single production line can easily generate about 1 TB of data each week. Managing large-scale images while the factory is in operation is a challenging task. Therefore, image compression is necessary within the industrial field. A lossy image compression can be an effective solution to reduce image capacity, but the image quality degrades. In this case, JPEG compression and pixelation can occur.

These factors prevent the training of high-performance deep neural networks. Generally, deep neural networks are stable and robust as a result of learning numerous weights on large-scale training data. They can make high-performance predictions for new testing data through their complex hierarchies and rich representations of networks. However, several studies have proved that deep neural networks can be significantly degraded with minimal differences or data distribution contamination [35,36].

Manufacturing applications require defect detection performance close to 100% because only one minor defective part can cause a severe product defect. Therefore, we should very carefully employ deep learning models and review possible degradation factors in advance for industrial image data with contamination. In this study, the TDD-PCB [30] dataset is augmented with possible contamination in industrial sites based on [35], and we analyzed model performance under these challenges.

## 4. Methods of PCB Defect Detection

This section summarizes three methods using deep learning for PCB defect detection: (1) part image classification, (2) whole image understanding, and (3) direct defect detection. Selecting the appropriate method according to the manufacturing process situation (e.g., the volume of collected data, the purposes of the defect inspection systems, and possible image contamination during manufacturing) is critical. Detailed descriptions of each method are summarized in Table 4. (Anomaly detection [37] is a possible solution for PCB defect detection. However, we exclude the method from this study because anomaly detection is very different from the three methods (i.e., image classification, understanding, and defect detection) and has limited applications in industrial sites.)
**Part image classification** is a method that distinguishes classes of input images. It takes cropped image patches as model input and predicts the class of input cropped data. The prediction results include the class of electronic component images and whether they are defective. We can apply image classification when the location and size of each electronic part are known. This method focuses on classifying the electronic parts at a specified location.**Whole image understanding** examines whole PCB images and determines whether parts or circuits in the image are defective. Training for image understanding does not require the specified location of each part but only the images and their labels, so the training data collection is simple. This method is suitable to employ where the locations of defects at the PCB are not specified. It does not directly infer the location and size of the defective part but only determines whether the whole PCB image contains defective parts. Therefore, we must implement an additional algorithm, such as the class activation map (CAM) [38], to visualize predictions of the image understanding model.**Direct defect detection** receives the whole PCB image as input and predicts the location, size, and class of the defects. The PCB images and annotations on the location, size, and type of the defects are necessary to learn the defect detection model. If the defect can occur at an unspecified location rather than a specified location in the image, applying the defect detection method is advantageous.

The size of the defect is small compared with that of the PCB, and it is inefficient to perform deep learning using the whole image. Traditional AOI [39] systems have used two approaches to perform defect detection effectively. The first is to determine a defective candidate using a standard sample. Due to the nature of PCB manufacturing, which produces the same type of product in large quantities, it is often possible to obtain the same type of standard sample in advance as the test product. In this approach, a differential image between the test product and the standard sample can be used to determine a candidate location with a possible defect and then cut around the area for deep learning prediction [40]. The location of the defect is known; thus, the most suitable method is part image classification.

Second, if it is difficult to obtain a standard sample, the whole PCB image is divided equally into a specific size and inspected. In this approach, if it is necessary to identify the location of the defect accurately, direct defect detection can be applied. If only the presence or absence of defective parts in the PCB image is needed, image understanding can be applied. This study analyzes the defect detection performance for the three methods and reviews the possible performance degradation factors according to industrial data attributes mentioned in Section 3.

## 5. Experiments

### 5.1. Settings

This section analyzes how the quality and quantity of training data affect the performance of each PCB defect detection method. We used the TDD-PCB dataset [30], which was augmented through image flipping, resulting in a total of 21,336 images. The dataset is composed of six classes of defects, with each class containing approximately 3550 images. The proportion of training images among the total images was divided into the following five categories to check the effect of the number of training data: 80%, 50%, 20%, 10%, and 5%. For fairness in the performance evaluation, the image ratio and the number of defective classes were set to be the same for each type. Thus, 20% of the total images were separated in advance as testing images and were used equally for all experiments. Next, we generated contaminated PCB images that frequently occur in industrial sites to analyze how each contamination affects defect detection. To this end, we referred to a study [35] that proposed several image transformation methods to generate contaminated and distorted images that could occur in real-world environments. The 16 contamination types discussed in Section 3.3 were taken, except for weather-related contamination proposed in [35], which rarely occurs at PCB manufacturing sites. We set the parameter settings in the mentioned study [35] as the default for image contamination. Contamination is applied to test images, rather than training images, since it is unpredictable during manufacturing.

As introduced in Section 4, we validated three methods for PCB defect detection, and the experimental settings for them are as follows. For standard settings, we set batch sizes and epochs to 16 and 50 for model training, respectively. For data augmentation, we performed image flipping, which is simple but effective for improving model performance. We followed default settings for other hyper-parameters such as the learning rate, loss function, and dropout rate. For part image classification, we trained ResNet50 [41], a successful deep neural network structure for many computer vision applications. Unlike with other methods, we randomly generated nondefective electronic part images. As the part image classification test specified and cropped part images, nondefective part images are essential for a fair comparison with other methods. We measured classification accuracy for each part image to evaluate model performance. For whole image understanding, the same ResNet50 [41] was employed because it allows a clear comparison of methods for the same goal (i.e., defect detection). To evaluate model performance, we also measured the classification accuracy. To visualize the positions of estimated defects, we employed Grad-CAM [38]. For direct defect detection, we applied an object detector called YOLOv7 [42], which currently displays superior performance compared with other detectors. The mean average precision (mAP) was measured for performance evaluations. We reflected the pretrained model by ImageNet [13] as the initial weight parameters for ResNet50 [41]. For initial weights of YOLOv7 [42], we used the pretrained model by MSCOCO [14].

### 5.2. Defect Detection Performance According to Training Data Volume

We validated the defect detection performance for three methods according to the training data volume, as summarized in Table 5. Part image classification maintains classification performance better than other methods under a lack of training data. When 80% of learning data were used, the performance of the entire class was 98.8%, and even if only 5% of the learning data were used, the average performance was 96.8%. This method does not need to determine the location of the part but only to predict the class of cropped images. Therefore, maintaining classification performance with a small training data volume (5%) is possible. In particular, missing hole and nondefective classes, which are quite discriminating, prevent performance degradation compared with other classes because the shape and appearance of these classes are clearly distinguished from other circuit-related defects, such as mousebite, open, short, spurs, and spurious. Moreover, the classification accuracy for spur was the lowest at 93.2%, using only 5% of the training data.

Second, the degradation in classification accuracy can be clearly observed for whole image understanding, and the degradation step is the steepest among the three methods. Therefore, the image understanding method requires more data, and if the number of data is limited, we do not recommend this method for defect detection. The performance drop for the missing hole (part-related) defect is relatively gradual compared with other circuit-related defects. In this case, more circuit-related part images are required to mitigate the degradation of classification accuracy. In direct defect detection, despite the sufficient epochs for model training, the detection rate decreases rapidly as the number of training data decreases. Similar to other methods, the detection rate of the missing hole class is maintained while circuit-related defects decrease rapidly. To sum up, as you can see in Figure 4, whole image understanding was most sensitive to the lack of training data. On the other hand, the part image classification maintained its performance despite extreme learning data reduction. Model performance must be verified during the learning process according to the amount of training data and methods.

### 5.3. Defect Detection Performance According to Possible Contamination

The performance of the part image classification for the contaminated images is summarized in Figure 5a. Compared to other methods (i.e., whole image understanding and direct defect detection), it still displays stable performance for most contamination, but performance significantly declined for several contamination types, such as impulse noise, zoom blur, and saturation. Even in simple noise, the classification performance significantly decreased from 6% to 36%. Thus, deep neural networks are vulnerable to even slight perturbations. In addition, the model could not completely cope with blur and digital distortions, which significantly affect the shape or color of the PCB. Missing hole and positive classes still exhibited stable performance compared to other circuit-related classes for most distortion types but displayed unstable results for zoom blur or saturation changes. For part image classification, the average performance decreased by about 0.252% after contamination.

The results of the whole image understanding model are summarized in Figure 5b. Among various contamination types, zoom blur most significantly degrades image quality, and the resulting performance drop was also the most rapid in this method. Based on the experiment results, the most robust class for contamination was missing hole, and the most vulnerable class was mousebite. However, mousebite was the best-performing class with missing hole, achieving 99.4% in Table 5. Although the classification accuracy of the class is high, the method is not always robust to contamination. For whole image understanding, the average performance decreased by 0.264% after contamination. We visualized the difference in robustness through Grad-CAM [38], as presented in Figure 6. If the prediction is successful, the area of defective parts in Grad-CAM is also specifically determined (Figure 6a). Otherwise, the peak of the Grad-CAM is unclear, and there is no clear distinction (Figure 6b). Thus, whether the understanding of a specific part has been correctly performed can be effectively checked through the Grad-CAM.

Finally, the results of direct defect detection for contamination are summarized in Figure 5c. For zoom blur types whose images are severely damaged, the performance was very low for all classes, and the missing hole class performance was strong, as in the previous experiments. In the case of other circuit-related classes, except missing hole, the performance was significantly degraded when noise and geometric transformation were applied. The performance graph clearly confirms this tendency. As the defect detection method focuses on the appearance information of defects distinct from the background, performance degradation due to contamination is significant compared to other methods. For direct defect detection, mean average precision decreased by 0.165% after contamination. In Figure 7, we present the detection results for several challenging types of contamination. The detector exhibited a 100% detection rate for the images without contamination. However, after applying the contamination, it struggled to detect PCB defects, resulting in frequent false positives and negatives. Compared to the other two methodologies, it was robust to most contamination except for zoom blur and elastic transform. Data augmentation for each contamination type can be attempted in advance to prevent performance degradation. However, the data augmentation rate for each contamination type should be appropriately adjusted. If too much data augmentation occurs for contaminated images, the defect detection performance for noncontaminated images can be degraded.

## 6. Discussion on PCB Defect Detection

To apply deep learning models to industrial sites, data collection and processing, model learning, model application and verification, and system maintenance should be carefully considered. In addition, compatibility with existing facilities, such as the programmable logic controller [43] and manufacturing execution system [44], is also essential. While many problems can arise during the training of deep learning models, it is highly recommended to thoroughly review and address any potential issues before deploying the model for use in a manufacturing line. This study conducted experiments on various methods (Section 4) that can be employed to detect PCB defects. Various image contamination types (Section 3) that can occur in industrial sites during the process were also considered. This section discusses PCB defect detection by considering the challenges in manufacturing conditions.

First, the class and quantity of training data that significantly influence model performance should be reviewed in detail. The TDD-PCB [30] dataset used in this study provides six types of defective classes. Among them, the missing hole class occurs in electronic components (part-related class), and the remaining five classes occur in the circuit area (circuit-related class). All five defects can be regarded as one class if we determine whether a circuit region is defective. However, if it is necessary to determine the detailed type of each defect during the manufacturing process, it is relatively difficult to distinguish all circuit defects into five different defects due to their similar appearance. As proved in the experimental results, all three methods commonly perform stably for the missing hole class, even with insufficient training data or severe image contamination due to the difference in characteristics of the class from other defective classes. As much as possible, classes with highly similar appearances must be identified, and real data must be collected. Thus, we can investigate and review the training history of the initial model. Collecting defective data at industrial sites cannot be performed in a short period. A sufficient development period must be secured until the model is applied, and data must be continuously collected even after the model is applied. It is possible to augment data during preprocessing, before the model application. However, data augmentation is not significantly different from the collection data distribution; thus, too much data augmentation can drop the model.

Second, image contamination that can occur during system maintenance should be considered, and the model performance should be checked at all times. As summarized in Section 3.3, the PCB is manufactured through the chemical action of complex mechanical facilities in industrial sites, so the images are more likely to be contaminated than those in a general environment. Based on the experimental results in Section 5.3, model performance is poor, especially for blur-related contamination (e.g., defocus and zoom) caused by the movement of the camera or PCB. In particular, if the classes are similar in appearance, they are more vulnerable to such contamination. In addition, when few data are available, it is more difficult to cope with such contamination. Therefore, contamination factors that may occur in the manufacturing process must be avoided, and the facility must be managed steadily by monitoring model performance.

Third, appropriate models should be used according to the manufacturing environment and purpose. In this study, we categorized three applicable methods for PCB defect detection in Section 4. In summary, part image classification is suitable for inspecting parts and circuits at known locations. Part image classification cannot detect defects outside the specified inspection area. Compared to other methods, part image classification displayed stable performance, even with various disturbances, but it requires the most annotation (e.g., location and class of each part). Moreover, whole video understanding has the advantage of being able to train the model with less data annotation. Generally, this method is not highly accurate and does not specify the defect location. Thus, additional methods, such as Grad-CAM [38], should be used to specify the defect location. Whole video understanding can be used meaningfully by quickly applying it to assist workers during manufacturing. Direct defect detection directly detects the locations of defects. It requires additional annotation on the location and size of the defect. This method has the advantage of being able to detect a defect even when a defect occurs in an unspecified location in a PCB circuit. However, it is difficult to train a model that performs well in all situations because the method must predict the class, location, and size of defects at once. They are vulnerable to contamination, as indicated in Figure 5; therefore, we must carefully apply the model to industrial sites and continuously maintain the model.

The article describes the application of deep learning models to industrial sites, highlighting the importance of data collection, processing, model learning, application, and maintenance. The study focuses on PCB defect detection and discusses the challenges faced in manufacturing conditions. The authors suggest that appropriate models should be used based on the manufacturing environment and purpose and that factors such as data quality, image contamination, and model performance must be carefully considered. The study categorizes three applicable methods for PCB defect detection: part image classification, whole video understanding, and direct defect detection.

## 7. Conclusions and Future Works

In this study, we described the significance of data collection, processing, model learning, application, and maintenance in the application of deep learning models to industrial sites. By categorizing PCB defect detection approaches into three applicable methods—part image classification, whole video understanding, and direct defect detection—we provide a comprehensive overview of the challenges faced in manufacturing conditions. Our experimental results demonstrate the impact of various degradation factors, such as defect detection methods, data quality, and image contamination. Ultimately, we hope that this study provides readers with valuable guidelines for implementing defect detection methods using deep learning in a wide range of industrial fields.

Based on our study, there are a few areas that could be explored for the future work: (1) Developing new methods for PCB defect detection. While the study identified three applicable methods, there may be other methods that could be developed for detecting PCB defects that are more effective in certain manufacturing environments or for specific types of defects. (2) Optimizing the use of deep learning models in industrial sites. We emphasized the importance of carefully considering data collection, processing, model learning, application, and maintenance when using deep learning models. We can further focus on optimizing each of these areas to improve the effectiveness and efficiency of deep learning models in industrial applications. (3) Application of deep learning models in other industrial settings. While the study focused on PCB defect detection, the principles and methods discussed could be applied to other industrial settings where the detection of defects or anomalies is important.

## Figures and Tables

**Figure 1 sensors-23-02766-f001:**
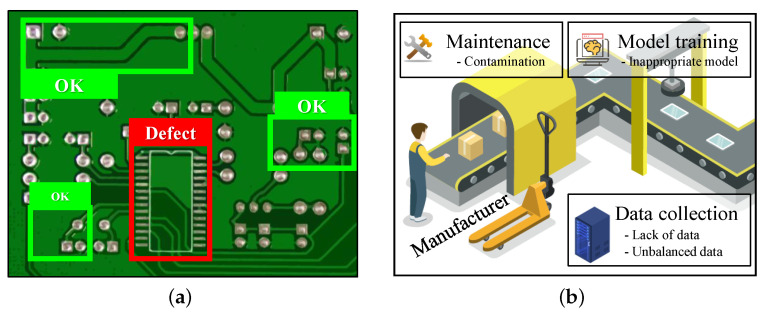
Results and issues of applying deep learning model for PCB defect detection. (**a**) Example of PCB defect detection. (**b**) Issues of applying a deep learning model for PCB manufacturing.

**Figure 2 sensors-23-02766-f002:**
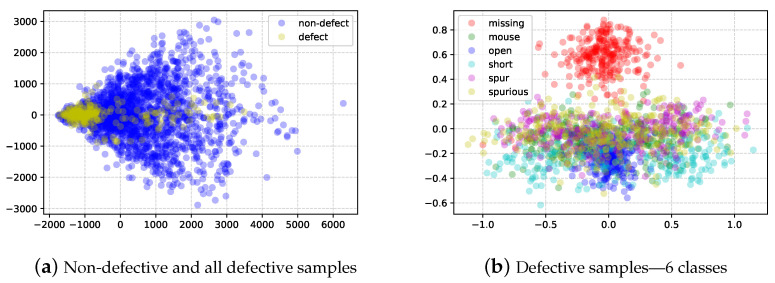
Distributions of TDD-PCB dataset [30]. (**a**) Imbalanced number of non-defective and defective samples. (**b**) Overlapped distributions of circuit-related defects (mousebite, open, short, spur, and spurious), whereas the distribution of a part-related defect (missing hole) is distinguishable.

**Figure 3 sensors-23-02766-f003:**
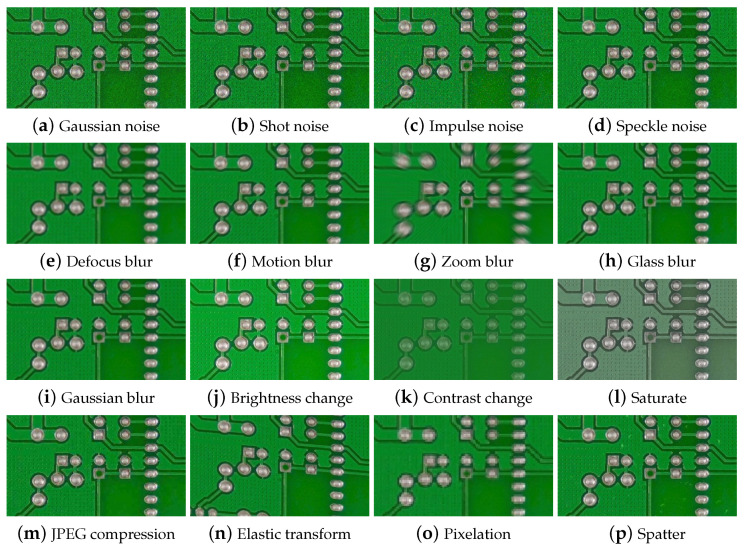
Example of deformation due to PCB image distortion.

**Figure 4 sensors-23-02766-f004:**
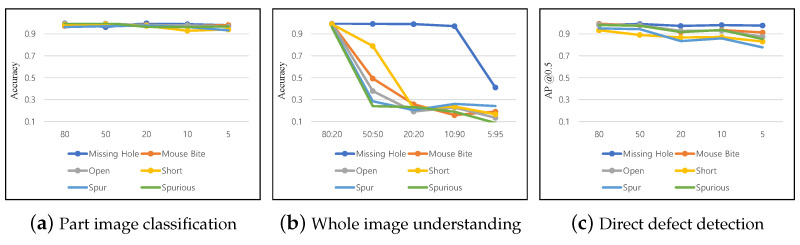
Performance graph of each methodology according to training data volume.

**Figure 5 sensors-23-02766-f005:**
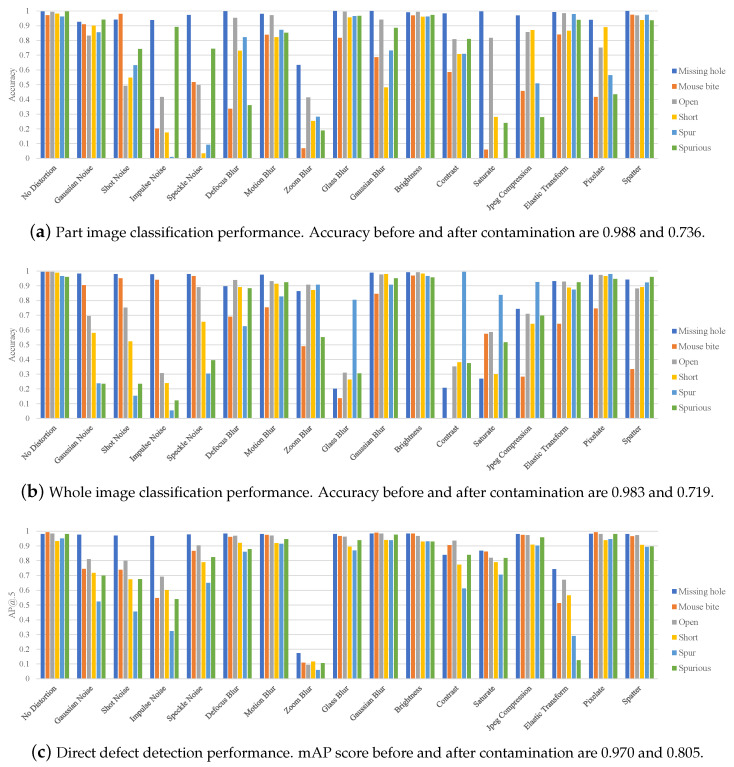
Defect detection model performance graphs due to various image contamination types. Best viewed in color.

**Figure 6 sensors-23-02766-f006:**
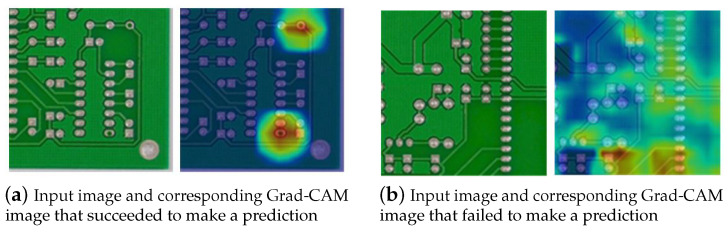
Examples of Grad-CAM [38] according to PCB circuit image understanding results.

**Figure 7 sensors-23-02766-f007:**
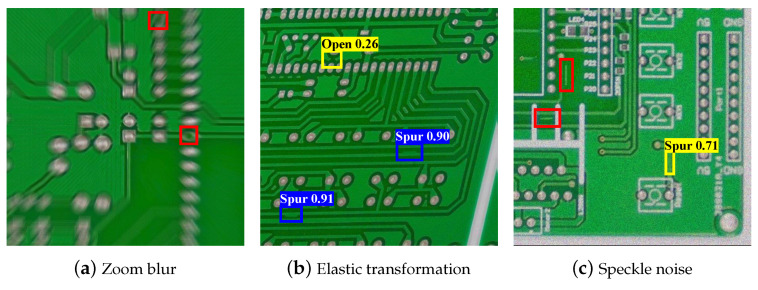
Examples of direct defect detection results under several types of contamination. Bounding boxes with red denote false negatives, blue denotes true positives, and yellow denotes false positives.

**Table 1 sensors-23-02766-t001:** Summary of public PCB datasets.

Dataset	Images	Cropped Images	Types of Defects	# of Defects	Types of Positives
PCB [29]	10	690	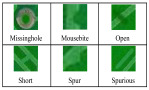	2070	Any circuits or parts except for defects
TDD-PCB [30]	10	10,668		21,336	
FICS-PCB [31]	31	400	None	None	IC Chip (3243) Capacitors (36,639) Resistors (33,182) Inductors (1292) Transistors (1398) Diodes (1593)
PCB DSLR [32]	165	849	None	None	IC Chip (9313)
PCB-METAL [33]	123	984	None	None	IC Chip (5844) Capacitors (3175) Resistors (2670) Inductors (542)

**Table 2 sensors-23-02766-t002:** Attributes of industrial data. We referred to [34] and summarized the primary attributes and their details.

Attribute	Descriptions
*Small amount*	– The amount of data collected is small. – Collecting industrial data usually takes a long time and many steps.
*Small object*	– PCB includes small and sophisticated electronic parts and circuits. – Close-up shots are required for detailed inspection of the individual small parts.
*Imbalance*	– The volume difference between non-defective and defective data can be huge. In general, it is more difficult to obtain data on defective parts. – Volume differences can also be large between defective parts.
*Fine-grained*	– PCB includes a variety of similar parts. – In addition, each part is further subdivided into various parts.
*Strong interference*	– Industrial process involves physicochemical reactions. – Considerable dust, steam particles, and vibrations can occur during manufacturing. – All the factors interfere with the clear imaging.
*Temporality*	– Industrial process consists of numerous sequential processes. – Long maintenance varies from the initial setup of the manufacturing facility.

**Table 3 sensors-23-02766-t003:** Possible image contamination due to four factors of industrial image acquisition environment.

Factor	Possible Contamination
Close-up imaging	• Defocus blur, Motion blur, Zoom blur, Gaussian blur • Elastic transform
Illumination changes	• Gaussian noise, Shot noise, Impulse noise, Speckle noise • Brightness • Contrast change • Saturate
Long-term maintenance	• Glass blur • Spatter
Systematic issues	• JPEG compression • Pixelation

**Table 4 sensors-23-02766-t004:** Summary of PCB defect detection methods.

	Part Image Classification	Whole Image Understanding	Direct Defect Detection
Training data	– Cropped images – Annotations: images	– Whole PCB images – Annotations: images	– Whole PCB images – Annotations: images, defect positions, and sizes
Test data	Cropped part image	PCB image	PCB image
Model prediction	Class of the part image	Class of PCB image	Defect location and class
Result examples	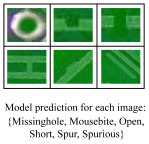	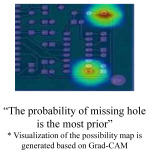	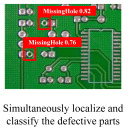

**Table 5 sensors-23-02766-t005:** Accuracy of three methodologies for PCB defect detection according to the amount of training data. Performance between 90% and 80% is in blue, 80–50% is in magenta, 50–20% is in red, and below 20% is in bold **red**. Best viewed in color.

**Accuracy of Part Image Classification (Model: ResNet50 [41])**								
**Data Ratio**	**Missinghole**	**Mousebite**	**Open**	**Short**	**Spur**	**Spurious**	**Non-Defective**	**Avg.**
80/20	0.996	0.972	0.993	0.982	0.962	0.997	0.998	0.988
50/50	0.962	0.980	0.994	0.988	0.969	0.999	0.961	0.975
20/80	0.997	0.983	0.980	0.972	0.976	0.963	0.964	0.974
10/90	0.988	0.972	0.979	0.928	0.962	0.962	0.995	0.975
5/95	0.979	0.980	0.966	0.940	0.932	0.967	0.983	0.968
**Accuracy of Whole Image Understanding (Model: ResNet50 [41])**								
**Data Ratio**	**Missinghole**	**Mousebite**	**Open**	**Short**	**Spur**	**Spurious**	**Non-Defective**	**Avg.**
80/20	0.994	0.994	0.994	0.988	0.965	0.960	-	0.983
50/50	0.991	0.493	0.379	0.789	0.285	0.241	-	0.532
20/80	0.989	0.258	**0.192**	0.222	0.205	**0.233**	-	0.354
10/90	0.970	**0.159**	0.229	0.237	0.262	**0.190**	-	0.344
5/95	0.411	**0.191**	**0.135**	**0.167**	0.242	**0.088**	-	0.207
**Mean Average Precision (mAP) of Direct Defect Detection (Model: YOLOv7 [42])**								
**Data Ratio**	**Missinghole**	**Mousebite**	**Open**	**Short**	**Spur**	**Spurious**	**Non-Defective**	**Avg**.
80/20	0.981	0.992	0.984	0.933	0.950	0.981	-	0.970
50/50	0.990	0.977	0.979	0.890	0.942	0.973	-	0.958
20/80	0.972	0.916	0.927	0.867	0.834	0.917	-	0.906
10/90	0.980	0.935	0.930	0.871	0.859	0.935	-	0.918
5/95	0.976	0.912	0.874	0.830	0.777	0.852	-	0.870

## Data Availability

Not applicable.

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
