# Peer review of "Analysis of Training Deep Learning Models for PCB Defect Detection"

_sensors, 2023, doi:10.3390/s23052766_

Round 1

Reviewer 1 Report

Automated inspection systemsare becoming essential for supporting automatically detect and decision-making and reliable PCB or any other manufacturing. These solutions become even more important for industrial development. In this sense, the manuscript proposes Analysis of Training Deep Learning Models for PCB Defect Detection. The paper is written in a straight and simple manner, although some details about the proposal are missing and some sections should be reorganized.  Also, some scientific and technical aspects deserve some specific comments.

Section 2: Related Studies should be reorganized in order to clarify by subsections the different sub scopes of the state of the art. For instance, Table 1 presents the list the studies focused on PCB defect detection. However, in the text is not clear why some works are not included in the table. Also, "Technique" of type " public PCB datasets" in table 1 is too unspecific and general. It should be more specific.  Authors should clearly claim the particular contributions of their work respect to the references found in the state of the art, i.e. the contributions and limitations of those respect to authors' proposal are not clear.

Reviewer would like to have 5 lines from the authors clearly describing what exactly they did in this work and what is their contribution.

Feature Selection FS can based on domain expert and/or FS techniques. The use of one or both should be explained, developed and argued in this section based on scientific references and/or specific FS method.

b.The details about the methodology of application of the selected algorithms and their hyperparameters are missing. They should be included and argued.

what  is the specific preprocessing methodology (data has been normalized if possible..? 

The entire Conclusions Section is full of trivialities and contains no findings, no pros and cons of this work and no future research directions.

Author Response

Thank you for your valuable comments. We reply all the comment in the attached file.

Reviewer 2 Report

Dear Authors, please, see the attached document. 

Author Response

(The authors gave the same response as above.)

Round 2

Reviewer 1 Report

This paper may now be accepted due to the fact that the authors have done a good job of accommodating all our questions